# A Submucosal Tumor-like Lesion of the Cervical Esophagus Similar to the Tonsillar Structures of Waldeyer’s Ring: A Case Report

**DOI:** 10.3390/medicina58121804

**Published:** 2022-12-07

**Authors:** Shibo Song, Xiaolong Feng, Xudong Liu, Guiqi Wang, Liyan Xue

**Affiliations:** 1Department of Endoscopy, National Cancer Center/National Clinical Research Center for Cancer/Cancer Hospital, Chinese Academy of Medical Sciences and Peking Union Medical College, Beijing 100021, China; songshibo1994@163.com (S.S.); liuxudong0401@163.com (X.L.); 2Department of Pathology, National Cancer Center/National Clinical Research Center for Cancer/Cancer Hospital, Chinese Academy of Medical Sciences and Peking Union Medical College, Beijing 100021, China; nccpathofxl@163.com

**Keywords:** submucosal tumor-like lesion, cervical esophagus, Waldeyer’s ring, endoscopic submucosal dissection

## Abstract

Esophageal submucosal tumors are rare, but their pathological types are diverse. In addition to the relatively common leiomyomas, some rare submucosal lesions are occasionally reported. Waldeyer’s ring is described as a unique subtype of mucosa-associated lymphoid tissue, located in the naso-oropharynx. Studies have reported that Waldeyer’s ring is the most common site of primary extranodal lymphoma in the head and neck. Interestingly, we encountered an esophageal submucosal tumor-like lesion similar to the tonsillar structures of Waldeyer’s ring. A 38-year-old man underwent esophagoscopy after experiencing swallowing discomfort for 3 months. A protruding submucosal mass with slightly rough mucosa was found at the cervical esophagus approximately 20 cm from the incisors. Considering the possibility of the coexistence of a submucosal tumor and a mucosal lesion, as well as the continuous symptoms of swallowing discomfort, the patient underwent endoscopic submucosal dissection. The lesion was removed en bloc. However, histology revealed a lesion similar to the tonsillar structure of Waldeyer’s ring between the lamina propria and submucosa. The patient was followed up for 6 months without recurrence or complaints. We report a new submucosal lesion and discuss its origin and treatment. Diagnostic ESD might be an effective option until its malignant potential is clarified.

## 1. Introduction

Submucosal tumors (SMTs) refer to the lesions originating from the submucosa and muscularis propria. The incidence rate of esophageal SMTs is relatively low, accounting for <1% of all esophageal tumors [1]. In addition to relatively common leiomyomas, some rare types of lesions have also been occasionally reported, such as gastrointestinal stromal tumor, fibroma, lipoma, etc. Waldeyer’s ring (WR) is a circular band of lymphoid tissue located at the opening of the digestive and respiratory tracts [2]. As a unique subtype of mucosa associated lymphoid tissue (MALT), WR has been reported to be the most common site of primary extranodal lymphoma in the head and neck [2]. Primary Waldeyer’s ring lymphoma (PWRL) accounts for about 5–10% of all lymphomas, and about 90% of PWRL are various types of non-Hodgkin lymphoma (NHL) [2,3]. Lesions similar to the tonsillar structure of WR outside the naso-orapharynx have not been reported. In this case, we encountered a rare submucosal lesion similar to the tonsillar structures of Waldeyer’s ring in the cervical esophagus of a 38-year-old man.

## 2. Case Presentation

A 38-year-old man underwent esophagoscopy after experiencing continuous swallowing discomfort for 3 months. A protruding submucosal mass with slightly rough mucosa was found at the cervical esophagus approximately 20 cm from the incisors (Figure 1A,B); the esophageal entrance is located about 18 cm from the incisors. Light staining and over staining spots can be seen on the mucosal surface after iodine staining (Figure 1C). These manifestations of the mucosal surface suggested an inflammatory lesion, but the possibility of intraepithelial neoplasia could not be ruled out. The patient had no history of gastroesophageal reflux disease (GERD), and there were no manifestations of GERD under endoscopy. Moreover, the patient denied the experience of choking on fish bones or other hard food. Endoscopic ultrasonography (EUS) showed a hypoechoic mass in the mucosal and submucosal layers, with a hyperechoic dot in its center, a clear boundary, and a maximum size of 2.9 × 5.2 mm, which suggested the possibility of small leiomyoma (Figure 1D). Physical examination, laboratory findings, and contrast-enhanced chest CT revealed no abnormality. A standard endoscopic biopsy found no tumorous lesions, only a small amount of papillary squamous epithelium with hyperkeratosis and parakeratosis, but the result may be a false negative from only a small amount of biopsy tissue. Considering the possibility of the coexistence of a submucosal mass and a malignant mucosal lesion, as well as the patient’s complaints about swallowing discomfort, the patient underwent endoscopic submucosal dissection (ESD) (Figure 2A–D). The lesion was removed en bloc.

Histology revealed a well-defined lesion between the lamina propria and submucosa (Figure 3A). It was composed of lymphoepithelial crypts lined by stratified squamous mucosa with infiltrating lymphocytes and histocytes, similar to the tonsillar structure of WR (Figure 3B). CD3+ T-cell zone and CD20+ B-cell-rich follicles were seen on immunohistochemistry (Figure 4A,B). Scattered PD-L1-positive lymphocytes and histocytes were observed. The epithelium was negative for PD-L1 (Figure 4C). P16 was negative in both lymphocytes and epithelium (Figure 4D).

The patient’s swallowing discomfort was significantly improved at 1 month, and the esophagoscopy at 6 months showed no evidence of recurrence.

## 3. Discussion

SMTs refer to the lesions originating from the submucosa and muscularis propria, whose surface is generally covered by normal mucosa [1]. The incidence rate of esophageal SMTs is relatively low, accounting for <1% of all esophageal tumors, and about 80% of them occur in the middle and lower esophagus [1,4]. The most common type of esophageal SMT is leiomyoma, accounting for about 70–80%, followed by gastrointestinal stromal tumor, fibroma, and lipoma [1]. In addition, some rare cases have also been reported in previous studies, such as granulosa cell tumor [5], lymphangioma [6], lymphoma [7], etc. It is worth noting that early esophageal squamous cell carcinoma can sometimes be encountered in the epithelium overlying a SMT [8]. EUS can show the origin and internal properties of SMTs, which is a useful technique for the diagnosis of SMTs [1]. However, the accuracy of EUS is significantly affected by observer experience [9]. The final diagnosis of SMTs requires histological confirmation, but common biopsy can only obtain a small amount of tissue from the mucosa and submucosa, which often cannot lead to an accurate diagnosis [1]. Although more than 90% of esophageal SMTs are benign, the possibility of malignancies still exists [1]. Guidelines recommended that the SMTs require resection if they cause obvious symptoms or the possibility of malignancy cannot be ruled out [10]. With the development of endoscopic techniques, endoscopic resection is reported to be more preferable to surgical resection for SMTs with a diameter less than approximately 20 mm in the muscular or submucosal layer [1,11].

WR was described as a unique subtype of MALT, located in the naso-oropharynx. The ring acts as a first line of defense against exogenous aggressors that enters the human body through the nasal and oral cavities. WR is composed of four tonsillar structures, including the pharyngeal tonsil located on the roof of the naso-oropharynx, tubal tonsils on each side, palatine tonsils located in the oropharynx, and lingual tonsils located on the back of the tongue. Histologically, WR is populated by lymphocytes such as B cells and T cells, as well as plasma cells, each of which encounter antigens through the epithelium. Except for the expression of B-cell and T-cell lineage markers in the lymphocytes of WR, PD-L1 was also expressed on most T cells. WR is an essential subtype of MALT, which may show a rarity of low-grade lymphomas and a high incidence of diffuse large B-cell lymphoma [3]. In addition, previous publications have shown that Hodgkin’s lymphoma can also involve WR [12,13]. As a consequence, early detection and removal of a lesion similar to WR is of great value and significance.

The patient underwent esophagoscopy for continuous swallowing discomfort. Although most small esophageal SMTs have no typical symptoms, the lesion located in the neck esophagus may cause obvious dysphagia [14]. In this case, we cannot rule out the possibility of the coexistence of an SMT and a mucosal lesion before surgery. In addition, the swallowing discomfort made the patient suffer a lot. After full communication, the patient underwent endoscopic submucosal dissection. The lesion was removed en bloc. Interestingly, the histology revealed a lesion similar to the tonsil of WR. Immunohistochemistry also showed similar characteristics to lymphoid tissue. WR has been reported to be the most common site of primary extranodal lymphoma in the head and neck. However, due to the atypical clinical presentation of lesions in WR, early diagnosis and treatment are challenging. The lesion was encountered in the cervical esophagus, which is different from the general location of WR. We speculated that it may be a lesion similar to WR, or an ectopic WR. If it is a lesion similar to WR, what is the risk of lymphoma? If it is an ectopic structure of WR, would it increase the risk of lymphoma when it appears in the abnormal site? The above assumptions need further study in the future. In addition, the slightly rough mucosa overlying the submucosal mass and the performance after iodine staining might result from the repeated friction of the protruding mucosa by food. In this case, we completely removed the lesion by ESD in time, avoiding the possibility of lymphoma in the lesion.

For such a rare lesion, the relevant management recommendations cannot be found at present, so we can refer to the guidelines for SMTs [10]. In this case, the lesion was encountered in the cervical esophagus, but sometimes it is difficult to clearly examine the lesions of the cervical esophagus under endoscopy. Therefore, in clinical practice, endoscopy physicians should be fully aware of the possibility of the missed diagnosis of cervical esophageal lesions, and fully consider the patient’s complaint symptoms. We should not neglect any swallowing discomfort, and should combine other imaging examinations such as EUS and CT when necessary. Considering that the lesion is located in the mucosa and submucosa, early complete endoscopic resection is a safe and effective choice for smaller lesions. However, the malignant potential of such lesions is unknown, so full communication with patients should be carried out before treatment. For lesions with a diameter of more than 20 mm, endoscopic resection is difficult, and surgical treatment can be considered. In summary, we suggest that treatment methods can be selected either endoscopically or surgically based on size, location, and patient preference.

To our knowledge, there were no reports of such lesions so far. Finally, because no specific tumor cells were seen in this lesion, we temporarily diagnosed the lesion as a submucosal tumor-like lesion similar to the tonsillar structure of WR.

## 4. Conclusions

This seems to be the first report of an esophageal submucosal tumor-like lesion similar to the tonsillar structures of WR. Whether it is a lesion similar to WR or an ectopic WR might require further exploration at the molecular or genetic level. Diagnostic ESD might be an effective option until this entity’s malignant potential is clarified.

## Figures and Tables

**Figure 1 medicina-58-01804-f001:**
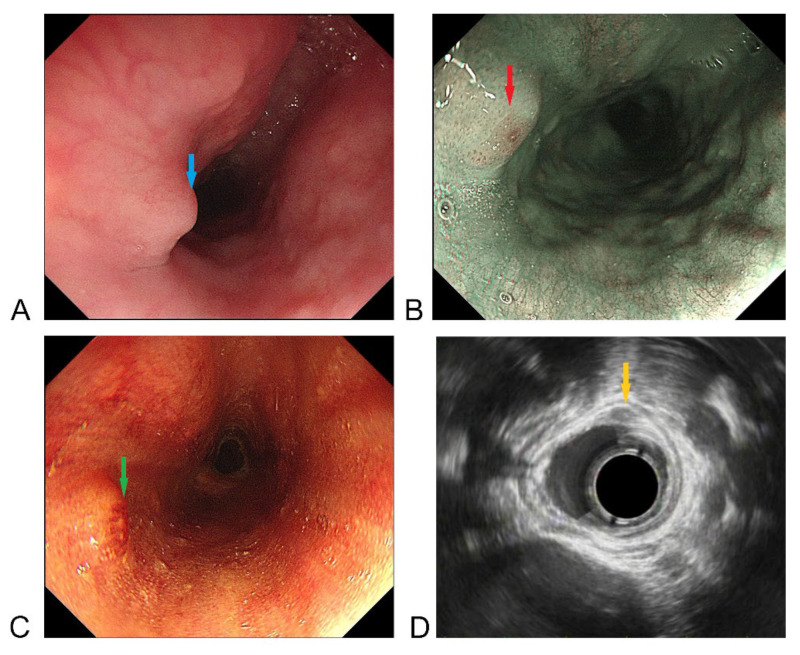
Endoscopic views of the esophageal tumor. (**A**) Endoscopy shows a protruding submucosal tumor-like lesion with slightly rough mucosa at the cervical esophagus about 20 cm from the incisors (blue arrow). (**B**) Narrow band imaging shows a barely visible intra-epithelial papillary capillary loop as brown dots (red arrow). (**C**) Light staining and over staining spots can be seen on the mucosal surface after iodine staining (green arrow). (**D**) Endoscopic ultrasonography shows a hypoechoic mass in the mucosal and submucosal layers, with dot hyperechoic in the center of the mass, clear boundary and maximum size of 2.9 × 5.2 mm (yellow arrow).

**Figure 2 medicina-58-01804-f002:**
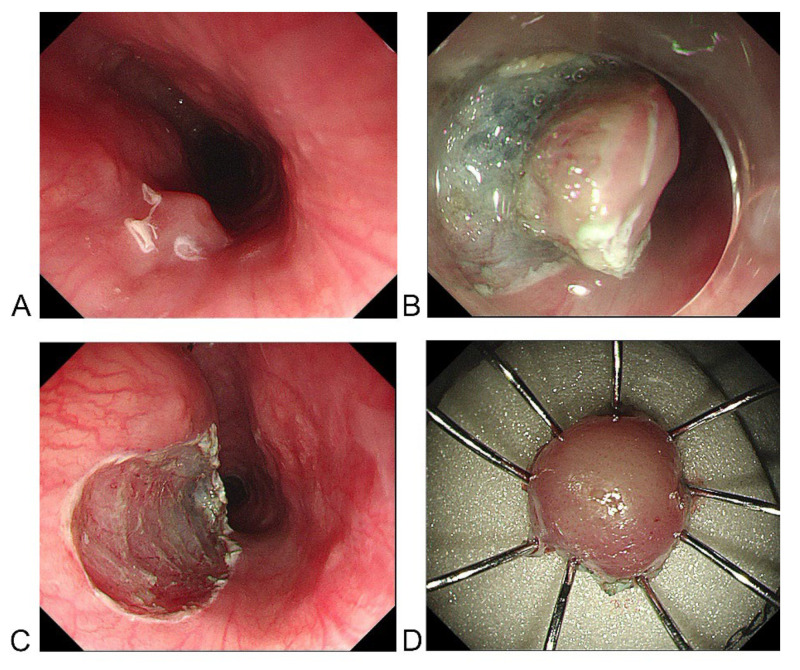
The process of endoscopic submucosal dissection: the lesion is removed en bloc. (**A**) The outer margin of the lesion. (**B**) Submucosal dissection. (**C**) Wound after dissection. (**D**) Specimen.

**Figure 3 medicina-58-01804-f003:**
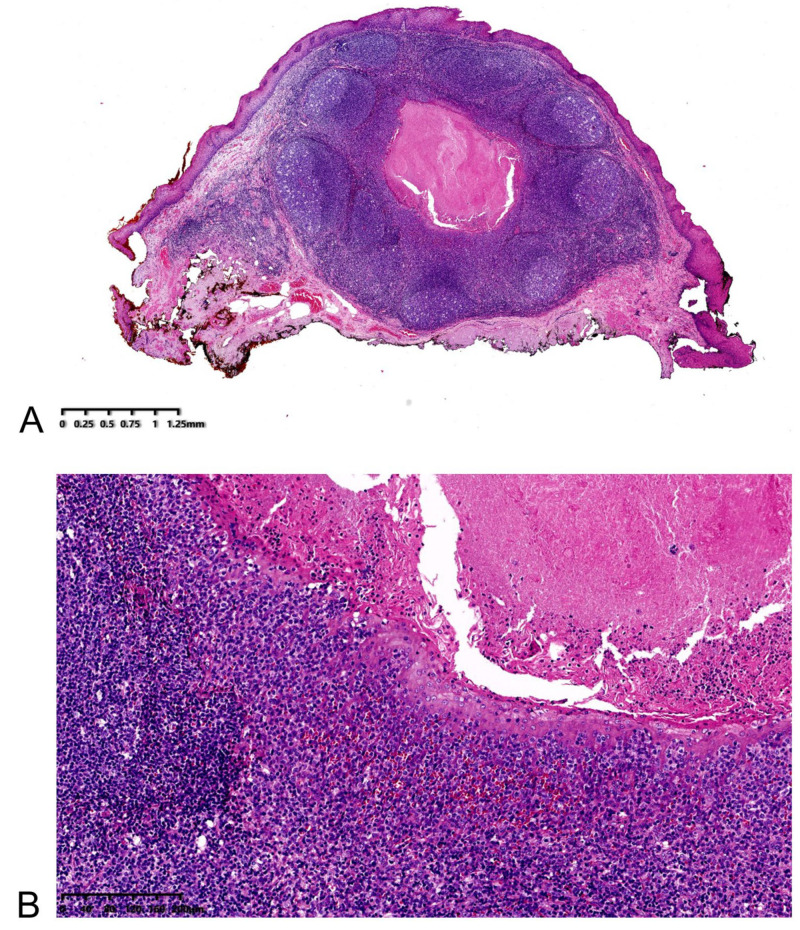
(**A**) At lower power, the lesion shows a well-defined location between the lamina propria and the submucosa. (**B**) High power displays lymphoepithelial crypts lined by stratified squamous mucosa with infiltrating lymphocytes and histocytes.

**Figure 4 medicina-58-01804-f004:**
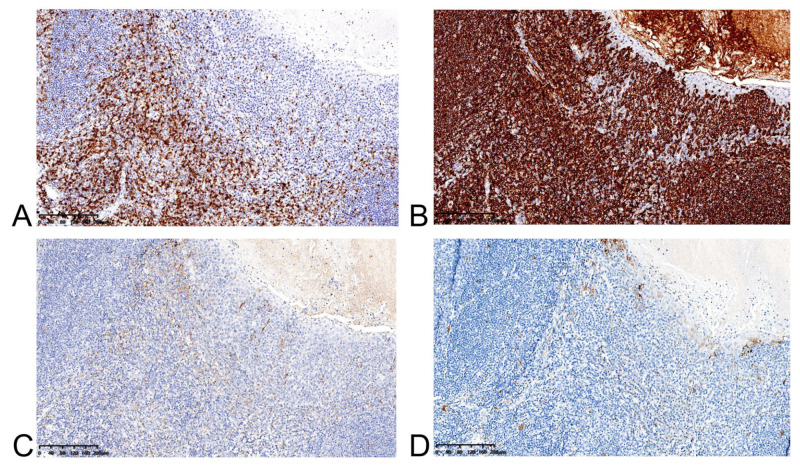
(**A**) CD3 staining highlights interfollicular T-cell-rich zone. (**B**) CD20 staining shows B-cell-rich follicles. Note the presence of more B cells within the lymphoepithelium (blue arrow). (**C**) PD-L1 staining shows a scattered positive in lymphocytes and histocytes, while epithelium is negative. (**D**) P16 is negative in both lymphocyte and epithelium.

## Data Availability

Not applicable.

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
