# Peer review of "A Submucosal Tumor-like Lesion of the Cervical Esophagus Similar to the Tonsillar Structures of Waldeyer’s Ring: A Case Report"

_medicina, 2022, doi:10.3390/medicina58121804_

Round 1

Reviewer 1 Report

This case report describes a rare submucosal lesion similar to the tonsillar structures of Waldeyer's ring in the cervical esophagus of a 38-year-old man. Lesions similar to the tonsillar structure of WR outside the naso-orapharynx are rare. 

Overall, the case report has merit and will be of interest to our readers. I have no major comments. However, being a rare lesion, I would like the authors to add a paragraph on the recommendations for the management of these lesions. 

Author Response

Thank you for your positive comments. We really appreciate your suggestions and advices. In response to your comments, we have modified our manuscript(the paragraph marked red). Detailed modifications are now listed hereinafter.

Added paragraph:

For such a rare lesion, the relevant management recommendations cannot be found at present, so we can refer to the guidelines for SMTs[10]. In the case, the lesion was encountered in the cervical esophagus, which is sometimes difficult to be clearly examined under endoscopy. Therefore, in clinical practice, endoscopy physicians should be fully aware of the possibility of missed diagnosis of cervical esophageal lesions, and fully consider the patient's complaint symptoms. We should not neglect any swallowing discomfort, and should combine other imaging examinations such as ultrasound and CT when necessary. Considering that the lesion is located in mucosa and submucosa, early complete endoscopic resection is a safe and effective choice for smaller lesions. However, the malignant potential of such lesions is unknown, so full communication with patients should be carried out before treatment. For lesions with a diameter of more than 20 mm, endoscopic resection is difficult, and surgical treatment can be considered. In one word, we suggest that treatment methods can be selected either endoscopically or surgically based on size, location, and patient preference.

Reviewer 2 Report

1. The authors presented the case, but if they present case history such as patient suffered from GERD, how long did he suffer from esophageal swallowing, etc..?

2. Did the patient take any medication?

3. Did the patient anytime identified with Helicobacter pylori?

Author Response

Thank you for your positive comments. We really appreciate your suggestions and advices.

  1. In response to your comments, we have modified our manuscript(the paragraph marked yellow). Detailed modifications are now listed hereinafter.

“A 38-year-old man underwent esophagoscopy for continuous swallowing discomfort for 3 months. “

“The patient had no history of gastroesophageal reflux disease (GERD), and there was no manifestation of GERD under endoscopy. Moreover, the patient denied the experience of choking on fish bones or other hard food.”

  1. The patient did not take any drugs orally except for the drugs used to eradicate Helicobacter pylori 1.5 months ago.

  1. The patient was found to be infected with Helicobacter pylori during physical examination in the local hospital 2.5 months ago. After the combined treatment of four drugs (rabeprazole sodium enteric coated tablets, amoxicillin capsules, clarithromycin tablets, colloidal bismuth tartrate capsules), it was retested negative by C13 breath test 1.5 months ago.

Reviewer 3 Report

I was pleased to review the article “A submucosal tumor-like lesion of the cervical esophagus similar to the tonsillar structures of Waldeyer's ring: a case report”.

The authors present a delicate subject, a rare pathology represented by the submucosal tumors, namely a particular case of an esophageal submucosal tumor.

These tumors are diagnosed, most of the time, in advanced locally stages and, when they are present in early stages, its can be misdiagnosed. This article presents a particular tumor that can be highlighted in the esophageal submucosal zone.

The title expresses clearly the content of the manuscript.  The abstract is a short and clear summary of the aims.

The introduction section clearly summarize the current state of the topic as well as clearly define the aim of the study.

In Case presentation, the authors logically explain and describe their findings. The images are representative, both macroscopic and microscopic. I suggest that the area of interest that has been described, to be identified by arrows in all the images.

In my opinion, the Discussion chapter contains useful information about the pathology discussed but, at the same time, it contains information about the presented case which , I think, belongs in the subchapter “case presentation” . I suggest  a revision of the Discussion and a supplement of information for Case presentation.

The authors concluded that this is the first report of such a case and the importance of EDS in diagnostic.

The authors cite the initial discoveries where suitable. The cited studies represent current knowledge.

Author Response

Thank you for your positive comments. We really appreciate your suggestions and advices. In response to your comments, we have modified our manuscript.

Firstly, the Figure 1 has been marked with arrows of different colors. However, for the figure of surgery (Figure 2), the descriptions are based on the entire surgical process. For pathological and immunohistochemical pictures (Figure 3 and 4), most of the descriptions are based on the performance of the whole section. It is meaningless to point out a small part with an arrow.

Secondly, we have revised the Discussion and supplied the information for Case presentation(the sentences marked blue).

These manifestations of the mucosal surface suggested an inflammatory lesion, but the possibility of intraepithelial neoplasia could not be ruled out. The patient had no history of gastroesophageal reflux disease (GERD), and there was no manifestation of GERD under endoscopy. Moreover, the patient denied the experience of choking on fish bones or other hard food.

“A standard endoscopic biopsy found no tumorous lesions, only a small amount of papillary squamous epithelium with hyperkeratosis and parakeratosis, but the result may be false negative for few biopsy tissues.

In the case, we cannot rule out the possibility of coexistence of a SMT and a mucosal lesion before surgery. And the swallowing discomfort made the patient suffer a lot. After full communication, the patient underwent endoscopic submucosal dissection. The lesion was removed en bloc. Interestingly, the histology revealed a lesion similar to the tonsil of WR. Immunohistochemistry also showed similar characteristics to lymphoid tissue.
